# Nonhuman Primates Are Protected against Marburg Virus Disease by Vaccination with a Vesicular Stomatitis Virus Vector-Based Vaccine Prepared under Conditions to Allow Advancement to Human Clinical Trials

**DOI:** 10.3390/vaccines10101582

**Published:** 2022-09-21

**Authors:** Christopher L. Cooper, Gavin Morrow, Maoli Yuan, John W. Coleman, Fuxiang Hou, Lucia Reiserova, Shui L. Li, Denise Wagner, Alexei Carpov, Olivia Wallace-Selman, Kristie Valentin, Yesle Choi, Aaron Wilson, Andrew Kilianski, Eddy Sayeed, Krystle N. Agans, Viktoriya Borisevich, Robert W. Cross, Thomas W. Geisbert, Mark B. Feinberg, Swati B. Gupta, Christopher L. Parks

**Affiliations:** 1IAVI, New York, NY 10004, USA; 2Galveston National Laboratory, University of Texas Medical Branch, Galveston, TX 77555, USA

**Keywords:** vesicular stomatitis virus, vaccine vector, Marburg virus vaccine, emerging infectious disease, filoviruses

## Abstract

Vaccines are needed to disrupt or prevent continued outbreaks of filoviruses in humans across Western and Central Africa, including outbreaks of Marburg virus (MARV). As part of a filovirus vaccine product development plan, it is important to investigate dose response early in preclinical development to identify the dose range that may be optimal for safety, immunogenicity, and efficacy, and perhaps demonstrate that using lower doses is feasible, which will improve product access. To determine the efficacious dose range for a manufacturing-ready live recombinant vesicular stomatitis virus vaccine vector (rVSV∆G-MARV-GP) encoding the MARV glycoprotein (GP), a dose-range study was conducted in cynomolgus macaques. Results showed that a single intramuscular injection with as little as 200 plaque-forming units (PFUs) was 100% efficacious against lethality and prevented development of viremia and clinical pathologies associated with MARV Angola infection. Across the vaccine doses tested, there was nearly a 2000-fold range of anti-MARV glycoprotein (GP) serum IgG titers with seroconversion detectable even at the lowest doses. Virus-neutralizing serum antibodies also were detected in animals vaccinated with the higher vaccine doses indicating that vaccination induced functional antibodies, but that the assay was a less sensitive indicator of seroconversion. Collectively, the data indicates that a relatively wide range of anti-GP serum IgG titers are observed in animals that are protected from disease implying that seroconversion is positively associated with efficacy, but that more extensive immunologic analyses on samples collected from our study as well as future preclinical studies will be valuable in identifying additional immune responses correlated with protection that can serve as markers to monitor in human trials needed to generate data that can support vaccine licensure in the future.

## 1. Introduction

Filoviruses are a major threat to global health and continue to impact the health security and geopolitical stability of Central and Western Africa. A major Ebola virus (EBOV) epidemic occurred in West Africa in 2013–2016 and has been followed by a concerning frequency of outbreaks in the Democratic Republic of the Congo and Guinea [1]. Outbreaks have been caused by EBOV recrudescence-related events in non-endemic parts of West Africa as well as recent zoonotic transmission in endemic regions [2,3]. Additional filoviruses remain endemic in animal reservoirs across Africa, including Marburg virus (MARV), Sudan virus (SUDV), and others that cause lethal hemorrhagic fevers in humans and have similar epidemic potential to EBOV [4].

Highlighting the risk of zoonotic transmission, modeling indicates that the geographic regions that might support transmission of MARV are quite extensive [5]. Moreover, a transmission event was detected for the first time in West Africa in a patient from Guinea with no travel history [6] and most recently in Ghana [7]. Outbreaks of filoviruses including MARV will continue to happen at an accelerated rate in the future with factors such as climate change, increased inter-continental travel, population growth, and zoonotic reservoir range expansion contributing to the likelihood of future disease transmission events [8]. In addition to the increasing risk of zoonotic transmission, it also is important to emphasize that the high case-fatality rate caused by MARV infection [9] makes MARV a substantial bioweapon threat [10,11].

Vaccination against filoviruses in response to natural or unnatural outbreaks and as a regular public health measure has the potential to help further control the health security threat to Africa as well as other regions that may be at risk of MARV importation or bioterrorism. The success of the EBOV vaccine produced by Merck Vaccines (rVSV∆G-ZEBOV-GP, marketed as ERVEBO^®^) has provided a strong rationale for efforts to develop other vaccines based on recombinant vesicular stomatitis virus (rVSV) vaccine vector technology [12,13,14]. The performance of ERVEBO^®^ in outbreak environments has clearly shown that the VSV-based technology has multiple features needed for development of other effective filovirus vaccines including, (1) acceptable safety and tolerability; (2) high efficacy after a single dose; and (3) rapid development of protective immunity [12,13,14,15,16].

Immune responses correlated with protection from filovirus infection are not completely understood [16,17,18,19]. Investigation of immune responses induced by ERVEBO^®^ in people indicates that anti-GP IgG serum titers and virus-neutralizing antibodies (nAbs) are positively correlated with protection [20]. Furthermore, monoclonal nAbs have been isolated from people vaccinated with ERVEBO^®^ further implying that these functional antibodies may be an important contributor to protection [21] although other IgG directed effector functions likely contribute [22]. In macaques vaccinated with different types of rVSV-based vaccines delivering MARV GP, anti-GP serum antibodies also have been associated with protection from disease provided by prophylactic vaccination [23] as well as vaccination soon before or shortly after MARV infection [24,25,26]. Notably, macaques exposed to MARV more than a year after vaccination with rVSV∆G-MARV-GP had persistent anti-GP binding titers and were completely protected even though nAb titers were low suggesting that additional antibody functions and immune effector cells are likely to be important contributors to the control of MARV infection and disease prevention [27].

Advancing a more detailed understanding of anti-filovirus GP immunity induced by rVSV-based vaccines is important, as it will be necessary to advance additional filovirus vaccine products based on this technology. Because of the sporadic nature of filovirus outbreaks, traditional human efficacy trials are not feasible. Thus, advancing new filovirus vaccines for use in people will need to follow alternative regulatory agency guidance like the US FDA Animal Rule [28] or Accelerated Approval Program [29], which can be pursued with strong data packages incorporating nonclinical efficacy data and identification surrogate markers for vaccine efficacy [30]. In the case of rVSV-based filovirus vaccines, these alternative regulatory agency mechanisms can be facilitated by the preclinical and clinical track record of rVSV∆G-ZEBOV-GP [12,13,14,31] and the extensive preclinical research conducted on MARV and other filovirus vaccines based on the rVSV∆G-ZEBOV-GP design [32,33,34].

In response to the first identified human MARV case in West Africa, the World Health Organization (WHO) convened filovirus expert groups comprised of infectious disease scientists, epidemiologists, public health experts, and vaccine developers to compose a research and development blueprint to enhance the WHO’s “Strategic Agenda for Filoviruses Research and Monitoring” (AFIRM) [35]. Central to this blueprint is understanding what experimental MARV vaccines are available and their state of development, and what preclinical data is available that supports their use in an outbreak situation. Furthermore, the blueprint will cover development of clinical trial approaches that can be utilized during a public health emergency due to a MARV outbreak. It is likely that ring vaccination strategies could be implemented as early as possible in response to an emergence of MARV or other filovirus threats, and that the availability of clinical trial material, as was the case for EBOV, would aid in the response to future filovirus outbreaks [36]. Without the availability of a vaccine prepared according to good manufacturing practices and ready for immediate use, the response to the 2013–2016 West African EBOV epidemic would have been much slower and had even more far-reaching consequences in terms of the toll on human lives and economics.

A MARV vaccine candidate (rVSV∆G-MARV-GP) based on the VSV technology used for ERVEBO^®^ has a strong preclinical track record of safety and efficacy [33,37,38,39,40,41]. To advance rVSV∆G-MARV-GP as a globally-accessible vaccine candidate for human use, we regenerated a recombinant vaccine strain using conditions that would support human vaccine development, after which we tested the immunogenicity and efficacy of the rVSV∆G-MARV-GP vaccine candidate in the cynomolgus macaque model for MARV disease [42,43]. The results showed that a single intramuscular (IM) injection of as little as 200 plaque-forming units (PFUs) of rVSV∆G-MARV-GP was 100% efficacious against Marburg disease and protected against development of MARV viremia after the animals were challenged with MARV Angola, which is known to be one of the more virulent MARV variants [44]. rVSV∆G-MARV-GP vaccination induced MARV GP-specific humoral responses that can be further interrogated to better understand correlates of protection and help provide an important bridge to future human safety and immunogenicity studies.

## 2. Materials and Methods

### 2.1. Cell Culture and Recombinant VSV

A research stock of rVSV∆G-MARV-GP encoding GP from the MARV Musoke variant [45], which was used in multiple earlier preclinical studies [40,41,46], and a VSV Indiana genomic plasmid was kindly provided by the Zoonotic Diseases and Special Pathogens Division of the National Microbiology Laboratory, Public Health Agency of Canada (PHAC). The nucleotide sequence of the rVSV∆G-MARV-GP genome was determined by Sanger sequencing as described before [47], after which a DNA fragment encoding the MARV GP gene was synthesized at GenScript (Piscataway, NJ, USA). The GP gene was then transferred into the VSV genomic plasmid as described earlier [47]. The nucleotide sequence of the new genomic clone was confirmed by Sanger sequencing and was propagated and purified using animal product-free medium and reagents.

Recovery of rVSV∆G-MARV-GP from plasmid DNA was initiated by electroporating Vero cells [47]. A cell bank used before that was qualified for human vaccine production [48] and was used at all stages of recombinant virus work (Figure 1). This cell bank was derived from the WHO working cell bank (WHO 10–87) deposited at the European Collection of Authenticated Cell Cultures (Vero [WHO], ECACC 88020401). Vero cells were cultured in Dulbecco’s modified Eagle medium (DMEM; Sigma Aldrich, Burlington, MA, USA) supplemented with 4 mM L-glutamine and 10% gamma-irradiated fetal bovine serum (FBS; Sigma Aldrich, Burlington, MA, USA). Electroporation was conducted using methods modified from those described before [47,49,50]. In brief, ~2.5 × 10^7^ cells were electroporated (low voltage mode, 3 pulses, 70 milliseconds, 140 V at 900 millisecond intervals) using a BTX830 apparatus (Harvard Apparatus, Holliston, MA, USA) and then were cultured at 37 °C in 5% CO_2_ and 85% humidity. Cell supernatant was harvested 3 days post-electroporation and used to infect Vero cell monolayers cultured in supplemented DMEM to amplify the new recombinant virus. Two days later, medium containing virus was collected, and stored at <60 °C. After confirming the rescued virus population had the expected genomic consensus sequence, three rounds of plaque isolation were performed.

Isolated virus plaques were picked from infected Vero cell monolayers overlaid with DMEM containing the supplements mentioned above with 2% FBS and 0.5% agarose (Lonza, Basel, Switzerland). Virus samples from multiple individual plaques were amplified in Vero cells, after which genomic sequence analysis was performed to identify lead candidates with the expected genome sequence. Lead candidates were then subjected to two additional rounds of plaque isolation after which selected candidates were used to infect Vero cells cultured in 5-layer Cell Stacks (Corning, Corning, NY, USA). Approximately 40 h after infection medium was harvested and clarified by low-speed centrifugation before being stored in aliquots at <−60 °C. Virus stocks were characterized using multiple assays to confirm expected genomic sequence, MARV GP expression, and no contaminants presence. The selected candidate was then designated as the pre-master virus seed (pre-MVS).

Vaccine material for preclinical studies derived from the pre-MVS was produced in Vero cell cultures and purified with a process based on tangential flow filtration (TFF). Briefly, Vero cells were seeded in a 5-layer Cell Stack with DMEM supplemented as described above and incubated for 72 h to achieve a monolayer that was near confluent. Before infection, the cell monolayer was washed two times with DMEM before adding 375 mL of Virus-Production Serum-Free Medium (VP-SFM; Thermo Fisher Scientific, Waltham, MA, USA) containing virus to achieve a multiplicity of infection (MOI) of 0.001. At 40 h after infection, medium containing virus was harvested and subsequently clarified by sequential filtration with a 1.2 µm filter (Sartorius, Gottingen, Germany) followed by a 0.8/0.45 µm depth filter (Pall Corporation, Port Washington, NY, USA). TFF was used to concentrate and further purify the virus using a 750 kDa hollow fiber membrane (Repligen Corporation, Waltham, MA, USA). This was followed by addition of MgCl_2_ (InVitrogen, Thermo Fisher Scientific, Waltham, MA, USA) to a final concentration of 1.5 mM and benzonase (200 U/mL; Sigma-Aldrich, Burlington, MA, USA) while continuing TFF for 30 min at room temperature. Buffer exchange was performed with 50 mM Tris-HCl, 150 mM NaCl, and 10% sucrose buffer, pH 8.0. Purified virus vaccine candidate was aliquoted and stored at <−80 °C. Figure 1 illustrates some of the characterization performed with the vaccine candidate, such as flow virometry, genome integrity, and MARV GP expression.

### 2.2. Flow-Virometry

Purified rVSV∆G-MARV-GP vaccine material was analyzed with an A60-MicroPLUS Apogee flow cytometer using highly purified Milli-Q water as sheath fluid. The sample was diluted 1:300 in sterile HBSS buffer and ran at 1.5 μL/minute with the autocycler set to 200,000 total events. A 405 nm violet laser was set to 150 mW and a Large-Angle Light Scatter detector was used to resolve virus peak profile.

### 2.3. Genome Integrity Analysis and Sequencing

Genome integrity of MARV GP was assessed by RT-PCR with Superscript IV One-Step RT-PCR System (Invitrogen, Waltham, MA, USA), whereas forward and reverse primers were in the VSV M and L genes, respectively. RT-PCR was performed at 60 °C for 10 min and 98 °C for 2 min, followed by 40 cycles of 98 °C for 10 s, 70 °C for 10 s, and 72 °C for 1.5 min with final extension at 72 °C for 5 min. A 2.9 Kb band was detected in an 0.8% agarose gel and excised for DNA extraction (Qiagen, Germantown, MD, USA). Sanger sequencing was performed with BigDye Terminator v3.1 Cycle Sequencing Kit (ThermoFisher Scientific, Waltham, MA, USA) and BigDye XTerminator™ Purification Kit (ThermoFisher Scientific, Waltham, MA, USA) with ABI 3500XL Genetic Analyzer (ThermoFisher Scientific, Waltham, MA, USA).

### 2.4. Analysis of GP Expression

Incorporation of GP in virions was monitored at multiple stages of production using Western blotting and methods similar to those described earlier [47]. Samples containing rVSV∆G-MARV-GP were denatured and separated using 4%–12% Bis-Tris denaturing polyacrylamide gels (Invitrogen, Waltham, MA, USA) and transferred with iBLOT2 system to nitrocellulose membranes (Invitrogen). To detect virion proteins, rabbit polyclonal anti-MARV GP (Cat. 0303-007, IBT Bioservices, Rockville, MD, USA) and anti-VSV N (produced in house [52]) were used as primary antibodies, and goat anti-rabbit HRP (Santa Cruz, Dallas, TX, USA) as secondary antibody. Signals were detected using SuperSignal West Femto Maximum Sensitivity (ThermoFisher Scientific, Waltham, MA, USA) and the ChemiDoc Imaging System (BioRad, Hercules, CA).

Flow cytometry was used to assess cell-surface expression of MARV GP and intracellular expression of VSV N. Adherent infected cells were detached from plates at 48 after infection by scraping them into a wash solution containing PBS supplemented with 0.5% BSA (PBS/BSA). Cell suspension was distributed into a 96 deep-well tissue culture plate before collection by low-speed centrifugation for 5 min at 860× *g*. For MARV GP staining on the cell surface, cells were initially resuspended in PBS/BSA containing mouse monoclonal anti-MARV GP (Cat. 0203-023, 5C1, IBT Bioservices, Rockville, MD, USA), rabbit polyclonal anti-MARV GP (Cat. 0303-007, IBT Bioservices, Rockville, MD, USA), or pan-filovirus-chimeric anti-GP mAb (Cat. 0200-003, IBT Bioservices, Rockville, MD, USA) at a final concentration of 1 μg/mL and incubated at room temperature for 25 min. Cells were collected by centrifugation, resuspended in PBS/BSA, and centrifugation was repeated to remove free anti-GP antibodies. Pelleted cells were resuspended in Cytofix/Cytoperm Solution (BD Biosciences, Franklin Lakes, NJ, USA) and incubated for 20 min at 4 °C in the dark. Permeabilized cells were collected by centrifugation and resuspended in Perm/Wash Buffer (BD Biosciences, Franklin Lakes, NJ, USA) before repeating centrifugation. To stain intracellular VSV N, cells were resuspended in Perm/Wash Buffer containing anti-VSV N mouse monoclonal antibody (Cat. EB0009,10G4, Kerafast, Boston, MA, USA) at 1 μg/mL final concentration, and incubated in the dark at room temperature for 25 min. Following incubation, the cells were collected and washed with Perm/Wash Solution as described above, cells were resuspended in a goat anti-mouse IgG1 or goat anti-Rabbit Alexa 555 and goat anti-mouse IgG2a Alexa 647 secondary antibody solution (Cats. A-21127, A-32732, A21241 respectively, ThermoFisher, Waltham, MA, USA) and incubated in the dark at room temperature for 25 min. Perm/Wash Buffer (BD Biosciences, Franklin Lakes, NJ, USA) was used for one wash step and resuspension of the cells. Flow cytometry was performed with BD SORP LSRII flow cytometer (BD Biosciences, Franklin Lakes, NJ, USA).

### 2.5. VSV∆G-MARV-GP Vaccination

Vaccination was performed in ABSL2 suites at the University of Texas Medical Branch (UTMB). The study design was approved by the UTMB Institutional Biosafety Committee (IACUC), and all animal research was conducted in compliance with the UTMB IACUC, Animal Welfare Act, and other federal statutes and regulations relating to animal care. The UTMB animal research facility is fully accredited by the Association for Assessment and Accreditation of Laboratory Animal Care.

This study contained six groups (*n* = four per group). Five groups were vaccinated with different doses of rVSV-MARV-GP ranging from 2 × 10^7^ PFUs down to 200 PFUs. The control group was vaccinated with a vaccine prepared from a similar VSV-based vaccine that expressed the Lassa virus glycoprotein (rVSV∆G-LASV-GPC [45,53]). Animals received one intramuscular (IM) injection in the quadriceps. Diluted unused vaccine material was back titered to confirm delivery of the targeted vaccine dose.

### 2.6. Analysis of Anti-MARV GP Serum IgG

Blood was drawn via peripheral venipuncture using serum separator tubes (Greiner Bio-One, Monroe, NC, USA) prior to, and on days 10 and 27 post vaccination. Serum was stored frozen (−20 °C) until analysis by indirect ELISA or plaque reduction assay to assess neutralizing antibody titers. Anti-MARV GP IgG endpoint titers were quantified using ELISA plates (96 half-well plates; Corning, Corning, NY, USA) coated overnight at 4 °C with a soluble form of recombinant MARV Angola GP (obtained from the US Department of Defense, Joint Program Executive Office for Chemical, Biological, Radiological and Nuclear Defense, CBRN-JPEO) diluted to 1 µg per ml in ELISA Coating Buffer (Biolegend, San Diego, CA, USA). After coating, the plates were blocked with blocking buffer (3% BSA in PBS with 0.05% Tween-20) for 1.5 h at 37 °C and then washed with 150 µL of wash buffer (PBS containing 0.05% Tween-20). Serum samples were then added in three-fold dilutions starting at a 1:100 dilution and incubated for 1 h at 37 °C. Following incubation, the plates were washed and incubated with anti-human IgG (H+L)-HRP (Jackson Immunoresearch, West Grove, PA, USA) at 1:6000 dilution for 1 h at 37 °C, washed again, developed using 1-Step Ultra TMB substrate (Thermo Fisher Scientific, Waltham, MA, USA), and stopped with 5 N Sulfuric acid (Thermo Fisher Scientific, Waltham, MA, USA) after 10 min. Plates were read within 30 min at 450 nm with a Molecular Devices (San Jose, CA, USA) VersaMax Microplate Reader using SoftMax Pro GxP Data Acquisition Software (version 5), Molecular Devices, San Jose, CA, USA. Serum from unvaccinated animals or serum taken prior to vaccination was included to determine assay background. Titers were defined as the serum dilution resulting in an absorbance >2-times the standard deviation of background wells. A commercially available anti-MARV GP antibody (IBT Bioservices, Rockville, MD, USA) served as a positive control in the assay. A Nonparametric ANOVA Test with Kruskal–Wallis and Dunn’s multiple comparison was completed on endpoint IgG titers using GraphPad Prism (version 9.0.0.), GraphPad Software, San Diego, CA, USA. Comparisons was completed between responses in the vaccine dose cohorts and the vector-control cohort. LOD was set at LOD/2 for statistical testing.

Virus-neutralizing anti-MARV GP serum antibodies (nAbs) were quantified using the vaccine material for rVSV∆G-MARV-GP (Musoke, see Section 2.1) as the target virus. Serum collected 27 days after vaccination was heat inactivated at 56 °C for 30 min, clarified by centrifugation at 9300× *g* for 10 min, then diluted serially from 1:20 to 1:327,680 and incubated for 1 h on a shake platform at 300 RPM at 37 °C with appropriate amounts of rVSV∆G-MARV-GP to produce about 100 plaques per well. After incubation, the serum-virus mixture was used to infect Vero cell monolayers in 96-well tissue culture plates. Following a 2-h incubation on shake platform at 300 RPM at 37 °C, the cells were overlaid with DMEM (ThermoFisher Scientific, Waltham, MA, USA) containing 1% FBS and 0.5% methylcellulose (ThermoFisher Scientific, Waltham, MA, USA). Plaques were allowed to develop for 44 h at 37 °C before the methylcellulose overlay was removed and cells were fixed with 7% *v*/*v* formaldehyde prepared in water (200 ul of per well) and incubated for 1 h at room temperature. Plaques were stained by adding 200 ul of crystal violet solution (0.33% in water) per well and incubating for 1 h at room temperature. Staining solution was removed, and the plaques were rinsed and dried before plaques were counted using a Cytation 5 Imager Gen5 3.08 software (Agilent, Santa Clara, CA, USA). The lowest serum dilution that decreased plaques by 50% or more was reported.

### 2.7. MARV Challenge Virus and Vaccine Efficacy

Challenge virus was prepared at UTMB using MARV Angola (200501379) isolated from an 8-month-old female patient in Uige, Angola. A challenge virus stock was developed from virus obtained from the CDC (CDC 810820) that was passaged twice in Vero E6 cells at UTMB [54]. On day 28 post-vaccination macaques were infected with 10^3^ PFUs by IM injection in the quadriceps. Animals were monitored daily and scored for MARV disease progression using a humane endpoint filovirus disease scoring sheet approved by the UTMB IACUC. The scoring changes measured from baseline included posture and activity level, attitude and behavior, food intake, respiration, and disease manifestations, such as visible rash, hemorrhage, ecchymosis, or flushed skin. Animals were also monitored for central nervous system abnormalities. A score of ≥10 indicated that an animal met the criteria for euthanasia. Blood was collected on days 4, 7, 10, 11, 13, 15, 21 and 28 after MARV challenge for evaluation of blood chemistries and quantification of infectious MARV. The UTMB facilities are accredited by the Association for Assessment and Accreditation of Laboratory Animal Care International and adhere to principles specified in the eighth edition of the Guide for the Care and Use of Laboratory Animals, National Research Council.

Infectious MARV in blood (viremia) was quantified using plasma collected from macaques and plaque assay [53,55]. Briefly, increasing ten-fold dilutions of plasma samples were allowed to infect Vero E6 monolayers (ATCC, Manassas, VA, USA) in duplicate wells (200 μL per well). The limit of detection from serum was 25 PFU/mL.

To monitor MARV genomes in blood (RNAemia) by RT-qPCR, RNA was isolated from whole blood with the viral RNA mini-kit (Qiagen, Germantown, MD, USA) using 100 μL of blood mixed with 600 μL of viral lysis buffer AVL [25]. Primers targeting the NP gene of MARV were used for real-time quantitative PCR (RT-qPCR) with a 6-carboxyfluorescein (6FAM)-5′-CCCATAAGGTCACCCTCTT-3′-6 carboxy-tetramethylrhodamine (TAMRA) probe. Thermocycler run settings were 50 °C for 10 min; 95 °C for 10 s; and 40 cycles of 95 °C for 10 s plus 59 °C for 30 s. Primers were synthesized by Integrated DNA Technologies and labeled probes were prepared by Life Technologies. MARV genomes in samples were calculated using a genome equivalent standard. The limit of detection for this assay is 1000 copies/mL.

## 3. Results

### 3.1. Generation of a rVSVΔG-MARV-GP to Support Human Vaccine Development

The rVSV∆G-MARV-GP [40,45] vaccine is based on a replication-competent chimeric virus design (Figure 1A) in which the gene encoding the natural VSV glycoprotein (G) is deleted (VSV∆G) and replaced with coding sequence for a functional glycoprotein from a heterologous virus [45,56]. The rVSV∆G-MARV-GP genomic clone was generated using the genomic plasmid for a lab-adapted VSV serotype Indiana [57,58] and GP coding sequence from the MARV Musoke variant [40,45]. To generate a rVSV∆G-MARV-GP strain suitable for human vaccine development, a new recombinant virus was regenerated from plasmid DNA as described in the Section 2.

We chose to advance VSV∆G-MARV-GP expressing the MARV Musoke GP as a candidate human vaccine for multiple important reasons, including preclinical evidence demonstrating: (1) a single vaccination with the VSV∆G-MARV-GP Musoke vaccine was shown to rapidly induce protective immunity in macaques [39,40,55]; (2) durable protective immunity was established following single-dose vaccination [27]; (3) vaccination with MARV Musoke GP induced cross-protective immunity against MARV Angola and RAVN [40,46]; (4) systemic vaccination with VSV∆G-MARV-GP provided protective immunity against both systemic and aerosol routes of MARV exposure [37]; and (5) the rVSV∆G-MARV-GP Musoke vaccine was evaluated in a neurovirulence study along with rVSV∆G-ZEBOV-GP and neither caused neurotoxicity after intrathalamic injection in macaques [41].

After a new rVSV∆G-MARV-GP Musoke strain was recovered from plasmid DNA, multiple clonal virus isolates were generated by conducting three rounds of plaque isolation (Section 2 and Figure 1A) while maintaining laboratory and documentation practices necessary to support human vaccine development. Several lead candidates subsequently were selected based on comparing the research virus [45] genomic nucleotide sequence to the new clonal virus isolates, and confirmation of GP expression by Western blotting and flow cytometry. Following amplification and banking of multiple seed stocks designated as candidate pre-master virus seeds (preMVSs), the viruses were analyzed using multiple assays described below to select the lead candidate that would advance to GMP master virus seed (MVS) production.

The assays and approach used to evaluate the pre-MVS candidates is summarized in Figure 1. Virus samples from the pre-MVS candidates were subjected to serial propagation in Vero cells to mimic manufacturing amplification to the MVS stage through final drug product manufacturing. During all stages of the preclinical vaccine production process, nanoflow cytometry was used to quantify virion particles (Figure 1B) and ensure that a single predominant peak of virions was detected, and that smaller particles or larger aggregates did not accumulate. Genetic stability was assessed using genomic nucleotide sequencing across the entire MARV GP insert to verify that no mutations within the coding sequence were developed during propagation and by an RT-PCR assay to assess the integrity of the GP gene utilizing primers flanking the GP insert like the ones shown in Figure 1C. Detection of an intact 2.9 Kb GP gene product (Figure 1C, Lane 2, positive control) was confirmed in both unpurified viral supernatants (Figure 1C, Lane 5) and virions purified using tangential flow filtration (TFF) (Figure 1C, Lane 7). Importantly, only RT-PCR products of the expected size were detectable, demonstrating that no GP gene insertion or deletion variants were present even in rare virus subpopulations. GP expression was confirmed by Western blot using whole-cell lysates or by analyzing purified virus for GP content during the purification process (Figure 1D). Additionally, flow cytometry was used to analyze GP expression on the surface of Vero cells infected with rVSV∆G-MARV-GP (Figure 1E).

To assess how pre-MVS candidates performed during a vaccine production process, virus derived from several pre-MVS candidates was amplified and purified to produce preclinical vaccine material using a lab-scale method that would mimic key aspects of a GMP manufacturing process. The vaccine material was prepared by infecting Vero cells, after which the virus was purified and concentrated using a scalable method based on TFF (see Section 2).

Based on the performance of the pre-MVS candidates in the laboratory assessment briefly summarized above, a lead pre-MVS was selected and was transferred to our GMP manufacturing partner. A qualified GMP master virus seed (MVS) has since been produced to support manufacturing. The preclinical vaccine material produced from the lead pre-MVS also was used for the preclinical vaccine efficacy study described below.

### 3.2. A Single Vaccination with a Wide Range of rVSVΔG-MARV-GP (Musoke) Doses Protects from MARV Angola Challenge

A preclinical research study (Figure 2A) was conducted in cynomolgus macaques to investigate the dose range over which rVSV∆G-MARV-GP was immunogenic and efficacious against challenge with a lethal dose of a low-passage MARV Angola. In brief, six groups of cynomolgus macaques (*n* = four per group) were vaccinated once by IM injection with doses ranging from 2 × 10^2^ to 2 × 10^7^ PFUs of purified rVSVΔG-MARV-GP. The control cohort received an IM injection with 2 × 10^7^ PFUs of another rVSV∆G-based vaccine encoding the Lassa virus glycoprotein (rVSVΔG-LASV-GPC) [53]. Following vaccination, samples were collected (Figure 2A) for measurement of immunogenicity and the macaques were then challenged 28 days post-vaccination with a lethal dose of MARV Angola and monitored for clinical signs of MARV disease (Figure 2B).

As illustrated in the plot in Figure 2B, all animals vaccinated with the control rVSV∆G-LASV-GPC vaccine developed clinical signs of MARV disease [42,43] and were euthanized on days 8 and 9 based on the humane endpoint clinical scoring sheet. The progression of MARV disease in the control animals indicated that an anti-VSV vector immune response did not interfere with MARV challenge. However, all animals vaccinated with rVSV∆G-MARV-GP (Figure 2A) survived (Figure 2B) and did not develop clinical features of MARV disease.

To assess how vaccination affected the presence of infectious MARV in peripheral blood following challenge, we quantified titers of MARV in plasma by plaque assay (Figure 2C). Determining the titers of infectious viruses is important since quantifying RNA copies by RT-qPCR does not measure viable virus progeny circulating in the blood, and importantly, an informative earlier preclinical study has shown that protection from EBOV disease progression in macaques was associated with maintaining an infectious titer in blood that was below a threshold titer of about 1 × 10^5^ tissue culture infectious dose 50 (TCID50) [59]. When we conducted the MARV plaque assay, viremia was detectable in samples collected on day 4 following challenge in all control animals and titers increased to 10^6^ PFUS per ml or more by day 7. At the time of euthanasia per protocol, titers were high at about 10^8^ PFUs/mL (Figure 2C). In contrast, infectious MARV was undetectable in serum at any time in animals vaccinated with high or low doses of rVSV∆G-MARV-GP.

MARV RNA copies also were evaluated by RT-qPCR using RNA extracted from whole blood (Figure A1). As expected, based on viral titers in plasma, RNA copies in control macaques were high and increased in parallel with titers of infectious virus (Figure 2C). Consistent with vaccination preventing detectable viremia, MARV RNA was detectable in just 5 of 20 vaccinated animals at any timepoint analyzed after challenge, and if RNA was detected, it was transient (Figure A1). For example, two animals in the group vaccinated with the lowest dose of 2 × 10^2^ PFUs of rVSV∆G-MARV-GP displayed a transient signal in the RT-qPCR assay. One of these animals was positive on day 4 and 7 with no additional detection out to the end of the study, while the second animal within this group was limited to detection at a single timepoint on day 7. Interestingly, we additionally observed a single timepoint (day 10 after challenge) when RNA copies were detectable in two animals from the group vaccinated with 2 × 10^4^ PFUs and one animal in the group vaccinated with 2 × 10^5^ PFUs (Figure A1). These RNA signals did not increase as would be expected if there was substantial MARV replication occurring in these vaccinated macaques. The presence of a low transient RT-qPCR signal at day 4 or 7 after challenge might be indicative of a MARV infection that was rapidly controlled or aborted resulting in little infectious virus being released into circulation, consistent with the viremia data in Figure 2C. The explanation for a transient RNA signal at day 10 in three animals is more speculative particularly since the RNA copies were detected late and did not correlate with the presence of infectious virus (Figure 2C), but perhaps this is related to clearance of the initial virus inoculum rather than progeny virions produced by active MARV replication. Overall, the analysis RNAemia was consistent with viremia, which both indicated that vaccination prevented significant MARV replication and release of viral progeny into the blood.

### 3.3. Humoral Immune Responses against MARV GP Induced by rVSV∆G-MARV-GP Vaccination

Serum was collected (Figure 2B) from all animals prior to and following vaccination to assess development of anti-GP serum IgG. Binding antibody titers at day 27 just prior to MARV challenge (Figure 3A) were quantified by ELISA against a soluble form of a recombinant Angola GP. Endpoint binding titers against the Angola GP were detectable in all animals indicating that a single vaccination, even with the lower doses, resulted in seroconversion. Humoral responses in animals vaccinated with rVSV∆G-MARV-GP displayed a wide range of IgG endpoint titers across the tested vaccine doses with an upper and lower limit of anti-MARV GP titers between 20,691–128, respectively. The highest IgG responses were detected in the group vaccinated with 2 × 10^7^ PFUs (mean titer 6974) and generally decreased in proportion to the reduced doses that were tested. The mean titers established with vaccine doses at or greater than 2 × 10^4^ PFUs were statistically significant when compared to the control NHP group vaccinated with VSV∆G-LASV-GPC. However, statistical inference was reduced as the endpoint binding titers reached the lower limit of detection (LOD = 100) for the ELISA assay in rVSV∆G-MARV-GP vaccine doses approaching 200–1000 PFUs (Figure 3A).

We also analyzed serum for virus neutralizing activity (Figure 3B) using a plaque reduction assay based on neutralization of rVSV∆G-MARV-GP (Musoke). The same type of VSV∆G-based assay has been used before to assess neutralizing antibodies against EBOV GP [20,60,61]. Neutralizing 50 titers (NT50) were generally low with mean NT50 responses ranging from 112–165 observed in the high-dose groups. However, this assay did detect neutralizing serum antibodies in nearly 90% of the NHPs (seven of eight animals) vaccinated with the higher vaccine doses (Figure 3B; 2 × 10^7^ and 2 × 10^5^ PFUs). Neutralizing serum antibodies also were detected in some animals vaccinated with lower doses, but low pre-vaccination background neutralization activity also was observed in some of the macaques from these groups. Nonetheless, three NHPs (NHP 9, 15, and 18) in the lower vaccine doses with detectable neutralizing responses also had the highest serum IgG titers within their respective cohorts.

## 4. Discussion

The replication-competent rVSV∆G-MARV-GP vaccine has been shown to be safe and highly efficacious in earlier preclinical studies [27,33,37,38,39,40,41,46,55], and we have developed a new recombinant virus and pre-MVS under conditions that will support production of vaccine material for use in human trials. Using this new rVSV∆G-MARV-GP, we have replicated the previous results showing that a single-dose of around 2 × 10^7^ PFUs is 100% efficacious and have extended the earlier data by showing that potent protection is achieved using much lower doses in cynomolgus macaques, a well-characterized filovirus disease model [42,43,62,63]. We found that a single dose of as little as 200 PFUs of rVSV∆G-MARV-GP (Musoke) prevented development of clinical signs of MARV disease (Figure 3A) following challenge with the more virulent Angola MARV variant [44].

Our data showing that low doses of rVSV∆G-MARV-GP are efficacious is consistent with an earlier study showing that very low doses of an rVSV∆G-EBOV-GP (Makona isolate GP) vaccine could prevent EBOV disease in cynomolgus macaques [59]. In this study, Marzi et al. showed that as little as 10 PFUs of rVSV∆G-EBOV-GP could protect from lethal disease and that anti-GP binding titers induced by vaccination were quite similar across doses from 10^7^ down to 10 PFUs. Moreover, following challenge in protected animals, they observed a rise in anti-GP serum titers suggesting that exposure to EBOV challenge virus elicited a rapid B cell response. Taken together, these data suggest that rVSV∆G-EBOV-GP administration followed by enough vaccine virus infection and replication to result in seroconversion established humoral immunity that played a central role in protection. The ability of a wide range of rVSV∆G-MARV-GP doses to similarly induce anti-GP seroconversion and protect macaques from lethal MARV challenge (Figure 2B and Figure 3A) lends additional support to this interpretation. Interestingly, although rVSV∆G-EBOV-GP and rVSV∆G-MARV-GP have a broad efficacious dose range in macaques, it is unclear if humans would be similarly protected by vaccination with very low doses. Human data from clinical studies with ERVEBO^®^ indicate that there is a decline in anti-GP ELISA titers at doses of 3 × 10^5^ PFUs or less [64,65,66,67], although seroconversion was still detectable suggesting that lower doses might be effective in people.

In addition to vaccination preventing development of MARV disease, we found that infectious MARV in the blood was undetectable by plaque assay in vaccinated animals. This indicates that immunity induced by rVSV∆G-MARV-GP provided a very effective barrier to development of viremia, which is consistent with earlier reports [27]. This is an important finding as the earlier preclinical investigation of dose and efficacy conducted by Marzi et al. with rVSV∆G-EBOV-GP indicated that immune control of EBOV replication that held viremia below a threshold of about 1 × 10^5^ TCID50 per ml was key to preventing disease progression [59].

As discussed above, it is reasonable to hypothesize that the antibody response is an important contributor to the efficacy of rVSV∆G-EBOV-GP and rVSV∆G-MARV-GP, but the protective mechanisms remain poorly understood. Studies in people vaccinated with rVSV∆G-ZEBOV-GP indicate that total serum IgG titers as well as neutralizing antibody titers against the vaccine virus provide some of the stronger correlates of protection in humans [20]. The potential importance of vaccination inducing functional antibodies might be emphasized by a study showing that antibodies with direct virus-neutralizing activity have been isolated from people vaccinated with rVSV∆G-ZEBOV-GP [21,68], but it is also likely that antibodies capable of mediating Fc-directed innate immune effector functions also play a role in protection from EBOV disease [22]. Thus, it was encouraging that we were able to detect direct virus neutralization activity specific to the MARV GP in serum from macaques vaccinated with the higher doses, and it will be informative to assess the functional properties of the anti-MARV anti-sera in more detail particularly since innate immune effector functions mediated by the antibody Fc domain are thought to play important roles in protection from MARV disease [69] and it is known that neutralizing monoclonal antibodies specific for the MARV GP are protective [70,71].

Better definition of markers associated with protection in macaques will be valuable, as they can be transitioned for use in human immunogenicity and safety trials for monitoring and comparing rVSV∆G-MARV-GP performance. Defined immunologic markers for use in human trials will be needed to generate data to support vaccine licensure using regulatory agency guidance for countermeasures that cannot be studied in traditional efficacy trials [28,29,30]. Moreover, identification of immunologic markers that allow a more detailed comparison of immune responses produced by rVSV∆G -MARV-GP in macaques to the immune responses induced by ERVEBO^®^ in humans and macaques will provide an important comparator for bridging rVSV∆G -MARV-GP performance to a highly effective human vaccine based on the same rVSV vaccine technology [30].

In summary, we have advanced a rVSV∆G-MARV-GP vaccine candidate that will be utilized for GMP manufacturing to support a Phase 1 clinical trial. Moreover, the GMP vaccine material, here-in developed and characterized, can be available in the event of an outbreak of MARV and can be rapidly advanced for manufacturing to generate stockpiles. Without available clinical trial material for rVSV∆G-ZEBOV-GP in 2014 to initiate clinical trials, it is unlikely that the international community would have had an efficacious EBOV vaccine to combat continued EBOV emergence and circulation in Central and Western Africa [13,72,73]. The recent MARV cases in Guinea and Ghana emphasize the need to make well characterized investigational vaccines available to better address emerging infectious disease pathogens.

## Figures and Tables

**Figure 1 vaccines-10-01582-f001:**
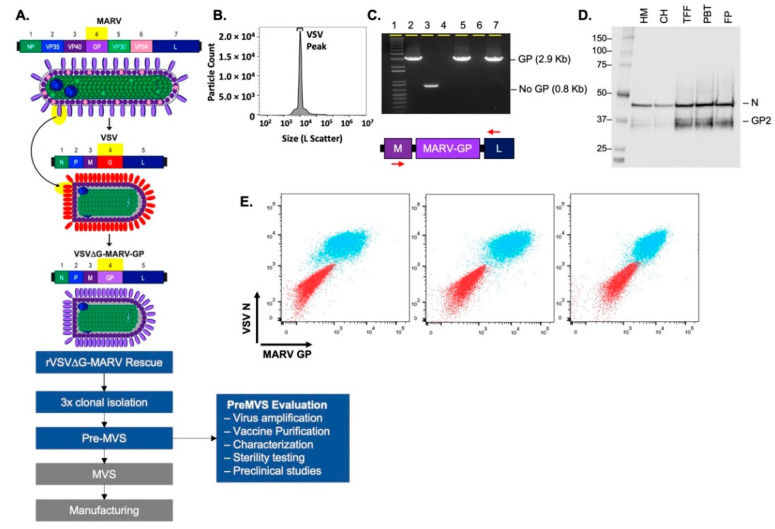
Generation and characterization of the rVSV∆G-MARV-GP vaccine for use in humans. (**A**) The schematic summarizes the rVSV∆G-MARV-GP chimeric virus design and critical work stages leading to cGMP MVS production and vaccine manufacturing. Following the regeneration of a new recombinant rVSV∆G-MARV-GP virus, 3 rounds of virus plaque isolation were performed prior to the generation and evaluation of a pre-MVS. (**B**) Nanoflow cytometry using an Apogee instrument was performed [51] with rVSV∆G-MARV-GP virions purified using tangential flow filtration (TFF). The graph shows particle counts (Y axis) and large-angle light scatter (X-axis). Total particles for the vaccine candidate were 6.7 × 10^10^ particles/mL, VSV virus peak (indicated) corresponded to 73.2% of the total particles. (**C**) GP gene integrity was evaluated by RT-PCR using primers specific for the VSV M and L genes that flank the GP insert. Analysis of PCR products by agarose gel electrophoresis showed that the expected 2.9 Kb product was amplified and that no larger or small products were detectable. Samples analyzed on the gel include: (1) 1 Kb ladder; (2) positive control VSV-MARV genomic plasmid template; (3) a negative control VSV genomic plasmid DNA (pVSV-G5) in which the G gene was moved to the 5′ terminus of the genome and no transcription unit is present between M and L [52]; (4) A negative control containing no template nucleic acid in RT-PCR reaction; (5) Medium harvest containing rVSV∆G-MARV-GP released by infected cells; (6) Empty well; (7) Purified rVSV∆G-MARV-GP. (**D**) Western blot analysis conducted using samples from various stages of virus purification. Blot probed with anti-GP rabbit polyclonal antisera that bound to MARV GP2 and rabbit poly-clonal anti-VSV-N. Analyzed samples included: Harvested medium (HM); clarified harvest (CH); product concentrated by TFF (TFF); product post-benzonase treatment (PBT); and final product (FP). (**E**) Flow cytometry was performed using Vero cells infected with purified rVSVΔG-MARV-GP vaccine material. Overlay dot plots displaying co-expression of MARV GP (x-axis) and VSV-N (y-axis) on uninfected VERO cells as controls (red population) and VERO cells 40 h post infection (MOI 0.001) with rVSV∆G-MARV-GP (blue population). GP was detected with three different anti-GP Abs (Pan-Filovirus anti-GP mAb (left panel), murine mAb 5C1 (middle panel), or rabbit anti-GP pAb (right panel)), and intracellular VSV-N was detected with murine mAb 10G4.

**Figure 2 vaccines-10-01582-f002:**
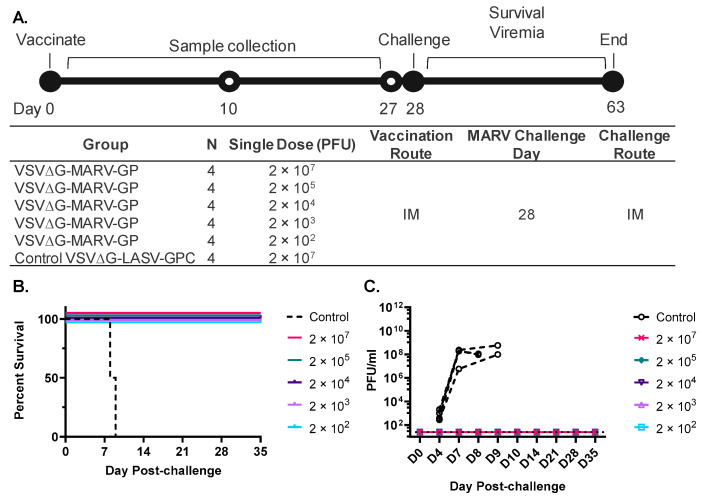
Design of rVSV∆G-MARV-GP preclinical dose-range efficacy study conducted in cynomolgus macaques. (**A**) Timing of activities during the course of the study are shown at the top and a table summarizing study design is shown below. The control rVSV∆G-based Lassa virus vaccine material (rVSV∆G-LASV-GPC; [45,53]) was produced from a new recombinant strain (IAVI unpublished). (**B**) Animal survival after MARV Angola challenge. (**C**) MARV viremia quantified by plaque assay at the indicated timepoints post viral challenge. Limit of detection (LOD) = 25 PFU/mL.

**Figure 3 vaccines-10-01582-f003:**
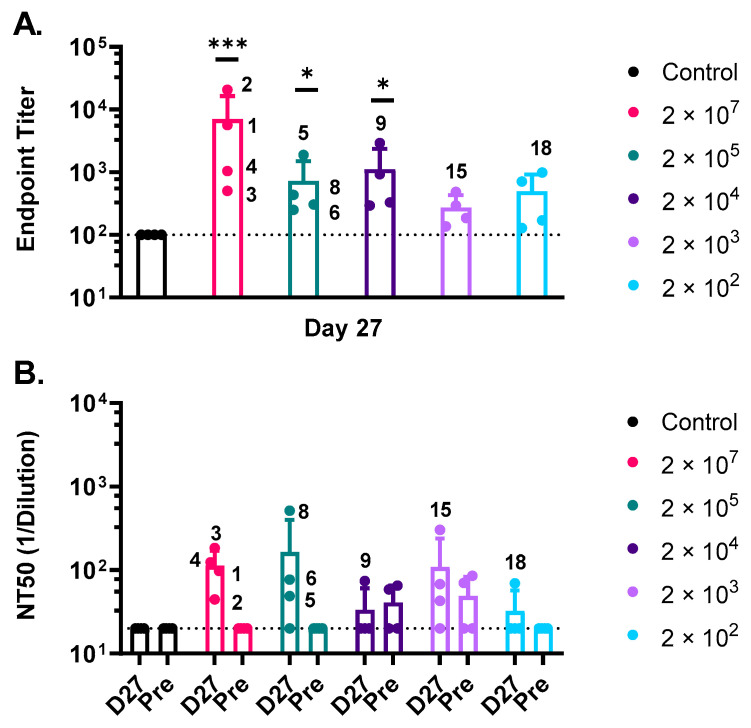
Characterization of the antibody response to vaccination. (**A**) ELISA was performed using plates coated with a soluble form of MARV GP Angola. Day 27 post-vaccination endpoint titers plotted as mean and standard deviation with animal numbers corresponding to high responders are shown in the graph. Vaccine dose in PFUs is included at the right side of the graph. The dashed line represents the lower detection limit of assay of 100. Statistical analysis was performed comparing vaccine cohort responses to vector control vaccinated animals using an ANOVA multiple comparison (*n* = 4, * *p* ≤ 0.05, *** *p* ≤ 0.001). (**B**) Serum dilution of pre-immune (Pre) and Day 27 (D27) post-vaccination at which rVSV∆G-MARV-GP (Musoke) plaque numbers were reduced by 50% (Neutralization titer 50 or NT 50) is plotted. The dashed line represents the lower detection limit of assay, 20. *N* = 4, mean and standard deviation.

## Data Availability

The data presented in this study are available within the article and supplementary material (Appendix A).

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
