# Peer review of "Nonhuman Primates Are Protected against Marburg Virus Disease by Vaccination with a Vesicular Stomatitis Virus Vector-Based Vaccine Prepared under Conditions to Allow Advancement to Human Clinical Trials"

_vaccines, 2022, doi:10.3390/vaccines10101582_

Round 1
Reviewer 1 Report
The manuscript by Cooper et al., entitled "Vaccination of Nonhuman Primates against Marburg Virus Disease using a Vesicular Stomatitis Virus vectored Vaccine – Implications for Dose sparing Strategies and Filovirus Outbreak Preparedness” describes an NHP efficacy study using research-grade rVSV MARV vaccine made at lab-scale using the vaccine virus construct to be used in GMP manufacturing. The authors report 100% survival of all NHP vaccinated with the rVSV vaccine candidate at all vaccine doses evaluated (2x10^7 to 2x10^2 pfu) and developed poor to no significant dose-dependent immune responses. The methods section is well-written with details regarding steps for derivation of the master virus cell bank for future process development in addition to the challenge materials and assays to be used in the NHP study. The animal study was reviewed by an Animal Ethics Committee.
The methods and outcomes sections suggest two study objectives: first study to derive a new virus cell bank for GMP manufacturing and 2) test their research-grade MARV vaccine made with the new vaccine construct in an NHP efficacy study to assess safety, immunogenicity and efficacy. The scientific value of the NHP study reported here as related to their MARV vaccine is high but this is not clearly articulated and becomes confusing when used to support discussions around dose sparing, stockpiling and filovirus outbreak preparedness. Establishing a safe and immunogenic vaccine dose in humans requires the testing of a GMP vaccine in humans. And any discussions of subsequent dose-sparing and stockpiling of a vaccine is precluded by human studies to obtain the knowledge of the safe and effective doses for the vaccine. All the doses evaluated by the authors suggested the vaccine was protective in the NHP model but whether any of these demonstrated NHP effective doses are optimal or even desired for use in humans remains unknown. Additionally, for most vaccines, adverse events show dose-threshold and dose responses, neither of which were readily evident in the study reported despite the wide range of doses tested. Identifying the minimal necessary dose to maximize the effectiveness – toxicity ratio in humans is not straightforward. It is also unlikely that many different doses (like those in this study) will ever be tested in large-scale human studies.
Regarding the NHP study related text, again the methods and outcomes sections are well-written and would benefit from a discussion of how outcomes met or did not meet the study objectives and why followed by an expanded discussion relating to the FDA Animal Rule. The Animal Rule states that FDA will rely on evidence from animal studies to provide substantial evidence of effectiveness only when their four criteria set forth in the Animal Rule are met. Of note is the last criteria, which is “the data or information on the kinetics and pharmacodynamics of the product or other relevant data or information, in animals and humans, allows selection of an effective dose in humans”, which was seemingly the objective of this NHP study. The discussion around the NHP study outcomes can be improved by expanding on outcomes in context of current stage of vaccine development and how it may inform the dose-schedule design to be used in their FIH clinical trial (and compare/contrast to what was done for ERVEBO vaccine); supports selection of the final dose of vaccine per vial in their GMP final drug product (and compare/contrast with ERVEBO); and informs next sets of methods to identify the vaccine’s correlate of protection and/or other markers to show clinical benefit in humans (and as informed by human immune and long term studies from the West African outbreak in addition to the ERVEBO clinical studies already included).
This Reviewer requests the manuscript be revised in title and text of the introduction and discussion to more accurately reflect the data reported and extrapolations it supports in the absence of human data relating to the author’s vaccine. Methods for the derivation and characterization of are provided in detail and outcomes nicely presented and could benefit from expanded discussion pertaining to these data supporting and informing both the NHP study and future manufacturing process development for clinical trials.
Typo on line 318
Typo on line 347
Author Response
We would like to thank the reviewer’s for their thoughtful and careful review. To address any comments or concerns we have provided the following revisions. Major revisions including rescoping of the introduction and conclusion were included to better present the benefits of our current body of work in the context of clinical relevance and are highlighted in blue in the revised manuscript. We have also provided additional insight on our vaccine characterization and observed in-vivo responses as they relate to correlates of filovirus protection and FDA vaccine licensure. The revisions requested by each reviewer that have been addressed are highlighted below. Thank you again for the thorough review and we are excited about communicating these results to the filovirus vaccine community.
All major changes in the text have been highlighted in Blue color in the revised manuscript. All other changes have been completed with track changes. Below are changes listed by line.
Lines 2-5: Title has been revised.
Lines 18-37: The abstract has been rewritten based on reviewers’ comments to refocus the report to the research findings and implications.
Line 67: Corrected abbreviation.
Lines 71-95: Revised introduction section to further highlight importance of Filoviral correlates of protection.
Lines 126-128: Removed sentence.
Line 233: Proofreading correction.
Line 273-277: Included statistical analysis for serum responses in Figure 3.
Lines 357-374: This section has been revised for clarity and based on reviewers’ comments.
Line 377: Proofreading correction.
Lines 408-410: Figure legend has been updated for flow cytometry data.
Line 422-423: Removal of sentence.
Line 426: Corrected Figure callout.
Line 434-435: Included LOD definition in Figure legend.
Line 438: Corrected euthanized days.
Line 449: Corrected Figure callout.
Lines 487-496: Revised and included additional text on immunogenicity results.
Line 500: Provided introduction and abbreviation for NT50.
Lines 500-508: Included additional text on immunogenicity results.
Line 509-510: Revised Figure 3 to highlight NHP identification and statistical significance.
Lines 512-520: Updated Figure legend for clarity.
Lines 537-549: Revised test in discussion on historical data from VSV-based studies.
Line 553: Additional text.
Lines 558-560: Revised text.
Lines 569-570: Removed text.
Line 575-597: Revised text for discussion
Reviewer 2 Report
Cooper at al. report an investigation on dose response of using VSV-vectored vaccine against Marburg virus (MARV) disease in NHP model. Authors performed extensive dose range study using single vaccination which showed as low as 200 PFU of viral vectored vaccine protected NHPs from lethal dose of MARV Angola variant. They report higher binding antibody and neutralizing antibody for higher dose of vaccine. Even though the neutralizing Ab was lower/reduced in lower dose vaccine, authors discussion few potential reasons in the discussion section. After successful ERVEBO vaccine (VSV-Ebola-GP) against Ebola virus during outbreak environment, this study has initiated an effort to develop vaccine based on VSV against another deadly filovirus-MARV. The dose sparing strategy suggested by the current study will be very useful when testing in humans for clinical studies. If low dose is as efficient in protecting against MARV disease in humans, in addition to decreasing cost of production it will decrease the cost burden for financially burdened countries where MARV disease is prevalent. Thank you for such descriptive materials and method section, this will help others replicate this study well if needed.
I would like to say Kudos to the authors for performing such an important study! Thank you.
I have minor concerns:
1. Line 325, did you mean Intraperitoneal injection?
2. Line 335, Flow cytometry data is shown in Fig 1E. Please change it in the result section. Please explain what blue and red population are showing in the fig 1E.
3. Line 347, performed
4. Fig 1B can be included earlier in the result as this result section felt it came all of a sudden and didn't go with the flow of the story.
5. Fig 1C, please include lane labeling in the figures (not only in the legends)
6. For Figures 2, I wished authors included the clinical signs measured rather than just stating that briefly. Also, if possible authors could include some cytokine response study (IL-6, INF-Alpha, TGF-Gamma, MCP-1 etc.) to show low activity is directly proportional to low dose (which is still protective). This would be a great result if it is that way.
7. I think authors can include the supplemental figure (RNAemia) as 2D.
8. Line 406, LOD?
9. For MARV RNA copies, what gene was used? I may have missed this in the manuscripts, if so please make it more evident.
10. Line 451, Fig 2 A not B.
11. In fig 3B, if authors can point/indicate which samples are what to show if the increase is from previously higher neutralizing sample would be beneficial for the reader.
12. If authors can elaborate/discuss the role of cellular immunity (especially T-cells) in the context of low dose vaccinated NHPs which were protected would be great.
Author Response
I have minor concerns:
- Line 325, did you mean Intraperitoneal injection? The use of parenteral has been changed to systemic for less confusion for the readers.
2. Line 335, Flow cytometry data is shown in Fig 1E. Please change it in the result section. Please explain what blue and red population are showing in the fig 1E. The data not shown refers to data obtained on candidate down selection. For simplicity, we have only provided detailed characterization of the lead pre-MVS candidate. However, Figure 1E flow cytometry data on the lead candidate has been clarified in the text and figure legends.
3. Line 347, performed. Thank you. We have corrected this proof-reading error.
4. Fig 1B can be included earlier in the result as this result section felt it came all of a sudden and didn't go with the flow of the story. We have introduced the Apogee Flow Virometry earlier in the results section for better clarity for the reader.
5. Fig 1C, please include lane labeling in the figures (not only in the legends). We thank you for your suggestions, we have updated the text for better clarity on this data.
6. For Figures 2, I wished authors included the clinical signs measured rather than just stating that briefly. Also, if possible authors could include some cytokine response study (IL-6, INF-Alpha, TGF-Gamma, MCP-1 etc.) to show low activity is directly proportional to low dose (which is still protective). This would be a great result if it is that way. We thank the reviewer for their insight and interest in additional data to further understand the differences across vaccine doses. Unfortunately, this data is not available at the current time. However, we have collected serum samples and PBMCs during the course of this study to capture both cytokine and transcriptional signatures associated with our rVSV∆G-MARV-GP vaccine. However, given the lengthy time and sources required to generate the accompanying dataset, this information is unfortunately out of the scope of the current manuscript. We look forward to sharing this additional information once available.
7. I think authors can include the supplemental figure (RNAemia) as 2D. We have elected to keep the RNAemia in the supplemental figure for reduced editing and formatting constraints.
8. Line 406, LOD? Thank you, we have provided the definition for this abbreviation on limit of detection.
9. For MARV RNA copies, what gene was used? I may have missed this in the manuscripts, if so please make it more evident. Primers targeting the NP gene of MARV was used for genome copy numbers. We have updated the text to highlight the gene targeted of this assay.
10. Line 451, Fig 2 A not B. We have updated the figure callouts. Thank you.
11. In fig 3B, if authors can point/indicate which samples are what to show if the increase is from previously higher neutralizing sample would be beneficial for the reader. We thank the review for their suggestion. We have updated Figure 3 to include individual animal identifications for tracking responses.
12. If authors can elaborate/discuss the role of cellular immunity (especially T-cells) in the context of low dose vaccinated NHPs which were protected would be great. Again, we thank the review for their suggestion, we have provided additional text around cellular immunity and correlates of protection in light of the range of protective humoral responses observed in our study.
Round 2
Reviewer 1 Report
The revised manuscript with needed supporting references, delivers a clear and convincing connection between the intent of the studies, data outcomes and the ensuing discussion.
However, please re-confirm your references for alignment to text - Several times i noted the references were one off - a couple examples are on Line 85 which shows reference [26] butthe text is supported by reference [27]; Line 541 shows a reference [63] when the text requires it to start at reference [64] etc.
Author Response
We have revised the edits to the manuscript word document. Endnote does not work well with the current template, so we will be diligent when reviewing page proofs to ensure everything has retained the correct annotations.